# Association of Gut Hormones and Microbiota with Vascular Dysfunction in Obesity

**DOI:** 10.3390/nu13020613

**Published:** 2021-02-13

**Authors:** Valentina Rovella, Giuseppe Rodia, Francesca Di Daniele, Carmine Cardillo, Umberto Campia, Annalisa Noce, Eleonora Candi, David Della-Morte, Manfredi Tesauro

**Affiliations:** 1Department of Systems Medicine, University of Rome “Tor Vergata”, Via Montpellier 1, 00133 Rome, Italy; dottorrodia@gmail.com (G.R.); francesca.didaniele@gmail.com (F.D.D.); annalisa.noce@uniroma2.it (A.N.); david.dellamorte@uniroma2.it (D.D.-M.); mtesauro@tiscali.it (M.T.); 2School of Applied Medical, Surgical Sciences, University of Rome Tor Vergata, Via Montpellier 1, 00133 Rome, Italy; 3Department of Internal Medicine, Università Cattolica del Sacro Cuore, Rome 00168, Italy; carmine.cardillo@unicatt.it; 4Vascular Medicine, Brigham and Women’s Hospital, Harvard Medical School, Boston, MA 02115, USA; ucampia@BWH.harward.edu; 5Department of Experimental Medicine, TOR, University of Rome “Tor Vergata”, Via Montpellier 1, 00133 Rome, Italy; candi@uniroma2.it; 6Biochemistry Laboratory, Istituto Dermopatico Immacolata (IDI-IRCCS), 00167 Rome, Italy; 7Department of Human Sciences and Quality of Life Promotion, San Raffaele Roma Open University, 00166 Rome, Italy

**Keywords:** obesity 1, endothelial dysfunction 2, metabolic syndrome 3, gut microbiota 4, gut hormones 5, Brain-Gut-Microbiome Axis 6

## Abstract

In the past few decades, obesity has reached pandemic proportions. Obesity is among the main risk factors for cardiovascular diseases, since chronic fat accumulation leads to dysfunction in vascular endothelium and to a precocious arterial stiffness. So far, not all the mechanisms linking adipose tissue and vascular reactivity have been explained. Recently, novel findings reported interesting pathological link between endothelial dysfunction with gut hormones and gut microbiota and energy homeostasis. These findings suggest an active role of gut secretome in regulating the mediators of vascular function, such as nitric oxide (NO) and endothelin-1 (ET-1) that need to be further investigated. Moreover, a central role of brain has been suggested as a main player in the regulation of the different factors and hormones beyond these complex mechanisms. The aim of the present review is to discuss the state of the art in this field, by focusing on the processes leading to endothelial dysfunction mediated by obesity and metabolic diseases, such as insulin resistance. The role of perivascular adipose tissue (PVAT), gut hormones, gut microbiota dysbiosis, and the CNS function in controlling satiety have been considered. Further understanding the crosstalk between these complex mechanisms will allow us to better design novel strategies for the prevention of obesity and its complications.

## 1. Introduction

Obesity is a major public health challenge of the 21st century, especially in urban settings, and in developing countries [1], where its incidence rises every year [2].

In 2016, the WHO estimated that more than 650 million adults were obese, and obese children and adolescent accounted for more than 340 million around the world. Since obesity related comorbidities, like diabetes and cardiovascular diseases, represent the leading causes of morbidity and mortality, there is an urgent global need to address the increase in obesity rate and its complications [3].

Even though several studies have been performed to understand all the factors underlying obesity and its related complication, so far, not all mechanisms are well understood. Novel, interesting theories highlight a pivotal role of gut flora and gut hormones in maintain the homeostasis of the central and peripheral organism axes controlling the metabolic function and accumulation of adipose tissue. Moreover, the same mechanisms have been also proposed to regulate the circulating factors involved in vascular homeostasis, further suggesting a multiple potential link with obesity and its complications.

Therefore, based on these considerations, investigating novel mechanisms involved in the regulation of fat accumulation, energy gain/expenditure, and appetite/satiety may be useful for the development of new treatments able to decelerate this emerging pandemic and its complications. 

This review aims to describe some of the mechanisms involved in the pathophysiology of obesity related vascular dysfunction, especially the pathological link between gut hormones, food intake, microbiome environment, and autosomal nervous system in the development of obesity related complications.

## 2. Methods

The current literature investigating obesity, endothelial dysfunction, gut hormones, and gut microbiota are analyzed and contextualized in this review. Specifically, research has been conducted on Medline (Pubmed). This review article analyzes studies (both in vivo and in vitro studies) published up to 2020.

## 3. Obesity and Vascular Complications

Obesity is a multifactorial disease, classified as a chronic and non-communicable disease and is characterized by an imbalance between calories consumed and energy expenditure [4]. The main risk factors associated to obesity are insulin resistance, age, sex, smoking, sedentary, genetics, among others [4]. All these factors predispose to an increase of chronic inflammatory state per se, which is typical of obesity, and to an increase of oxidative stress organic damage. 

Moreover, the chronic accumulation of fat leads to the deregulation of energetic storage homeostasis that in turn increase the circulating levels of saturated fatty acid and then glucose and lipids blood levels [5,6]. These events chronically lead to classical complications of obesity, which are dyslipidemia, cardiovascular disease, hypertension, atherosclerosis, and diabetes mellitus (T2DM) [7]. 

All these chronic diseases linked with obesity recognize a “fil rouge” in the double way of pathogenesis recognized as the lesion on the vascular bed that may become evident as the stiffness in arterial vessel or endothelial dysfunction. When these are evident as phenotypes of damage, they are already subclinical markers of cardiovascular disease, and major complications of obesity are already ahead. Understanding all the mechanisms and steps, even the less investigated, between interactions linking obesity, vascular risk factors, and associated vascular damage, is pivotal to prevent this pathological loop. It is important to also understand all the mechanisms leading to vascular damage that are endothelial independent.

## 4. Obesity, Metabolic Diseases, and Endothelial Dysfunction

The endothelium is an inner lining of the vessels composed by a singular layer of endothelial cells surrounded by smooth vascular muscle cells, interacting with each other to develop vascular response. In physiological conditions, the endothelium develops a tight control balance between vasodilation, through nitric oxide (NO) and prostacyclin production, and vasoconstriction by the regulation of endothelin 1 (ET-1) and thromboxane A2 activity [8,9,10].

The endothelium has either homeostatic and metabolic adaptation ability by regulating vascular response to several physiological stimuli like acetylcholine, bradykinin, catecholamines, and insulin or to injury factors, such as shear stress, temperature change, ischemia, reactive oxygen species (ROS), and Oxidized low-density lipoprotein (OxLDL), among others [11]. The vasoactive substances produced by the endothelium, like vascular endothelial growth factor (VEGF), play also a role in regulation of vascular growth and development [12].

As mentioned below, endothelial dysfunction is manly defined as an imbalance between NO bioavailability and increase in ET-1 vascular activity [13], and it may be triggered by several factors, such as aging, inflammation, oxidative stress, hypertension, and hyperglycemia, all typical of obesity and metabolic alteration [14]. Moreover, since endothelium is an active organ, besides NO and ET-1, in response to alteration in metabolic homeostasis (e.g., high circulating insulin levels), other factors are released to the final development of its dysfunction [15]. These factors are extracellular matrix proteins, hormones, growth factors, and several enzymes, such as prostacyclines [15].

Pathological conditions particularly cause several alterations of endothelial signaling leading to endothelial nitric oxide synthase (eNOS) uncoupling, soluble guanylyl cyclase desensitization (sGS), inactivation of prostacyclin synthase (CYP8A1), oxidative activation of the endothelin-1 system, and inactivation of NO by superoxide [16,17,18]. The final results of all these signaling pathway dysfunctions are an impairment in the endothelium-dependent vasodilation and increased proinflammatory and procoagulant activity [13]. This process is considered among the most important initial step of atherosclerosis towards cardiovascular diseases, and it is a fingerprint of obese-insulin resistant-diabetic subjects.

However, it is important to highlight that vascular dysfunction related to obesity and metabolic diseases, involves not only vasoreactivity in an endothelium-dependent manner (e.g., ach-NO mediated) but also other factors, such as co-impairment on vascular smooth muscle responsiveness to NO, as demonstrated by a reduced blood flow response to sodium nitroprusside, NO donor, in obese patients [19,20]. Interestingly, vascular improvement has been demonstrated to be independent of change in weight but correlates strongly with glucose levels. This suggests that obesity related endothelial dysfunction is manly associated to glucometabolic alterations rather than adipose tissue mass accumulation [21].

On the other hand, vascular homeostasis in obese subjects can be also altered by an increased activity of ET-1 system, as described following. ET-1 is considered the most powerful vasoconstrictor mediator, produced by endothelium; it is involved in pathophysiology mechanisms of endothelial vasomotor dysfunction and atherosclerotic plaque formation and progression [22]. Leg vascular responses to intra-arterial infusions of an ETA receptor antagonist (BQ123) restored impaired endothelial depend vasodilation in obese subjects, suggesting the important role of ET-1 in obesity related vascular dysfunction [23].

Furthermore, an enhanced ET-1 vasoconstriction tone has been shown in patients with metabolic syndrome. A significant higher vasodilator response to ETA antagonist accompanied with an impaired vasoconstriction response to the LMNA (NG-monomethyl-L-arginine), a NO synthase inhibitor, consistent with impaired NO bioavailability in obese vasculature, were present in these subjects [24]. Nevertheless, it is important to remark that ET-1 has also vasodilator effect. Its double effect is mediated by its receptors at vascular levels. In fact, through the activation of ETB Receptor, ET-1 induces the release of NO and other vascular release factors in the endothelium while by activating ETA Receptor it induces vasoconstriction [25].

We previously reported as obesity was associated with vascular damage independently by metabolic abnormalities underlying metabolic syndrome; indeed, patients with obesity but without abnormalities that define metabolic syndrome, have abnormal vascular reactivity, although their endothelial dysfunction is less pronounced than in patients with metabolic syndrome [19].

Metabolic syndrome is defined as a cluster of conditions that occur together, increasing the risk of cardiovascular diseases and T2DM [26]. These conditions include increased blood pressure, abnormal fasting blood glucose, excess body fat around the waist, and atherogenic dyslipidemia [27]. Several pathophysiological mechanisms may account for the development of this metabolic phenotype but insulin resistance and hyperinsulinemia seem to be the main players in this process [28]. Arterial endothelial dysfunction is the common denominator between insulin resistance, hypertension, and vascular damage [29]. ET-1 dependent vasoconstrictor tone has been shown to increase in patients affected by hypertension [30], hypercholesterolemia [31], T2DM [32,33], and metabolic syndrome, coexisting with an impaired NO bioactivity in the vasculature of these patients [24].

In the recent past, the insulin activity on endothelial cells has been widely studied [34,35]. At this moment, it is well known the role of insulin on activation of both vasodilator and vasoconstrictor pathways through the endothelial insulin receptor by different intracellular mediators, leading to vasodilation prevalence under insulin stimuli [36,37,38]. Physiologically, insulin induced-NO release in the skeletal muscle circulation could lead to an expansion of the capillary surface area in periphery in order to improve the delivery of nutrients, insulin, glucose and other metabolites to active tissues, thereby increasing insulin sensitivity [39]. Furthermore, insulin, besides anabolic function, to induce vasodilatation and reduce oxidative stress, promotes angiogenesis and proliferation of vascular smooth muscle cells [40]. An important role in the regulation of L-arginine transport has been also suggested as the pathway involved in insulin-mediated vasodilatation [41].

Interestingly, the vascular effect of insulin and impairment in its signaling have been studied in several vascular bed, such as carotids, renal vessels, brain cerebral arteries, and coronaries [37]. Specifically, coronary arteries show diminished effects of the vasoconstrictor insulin-dependent MAPK pathway in order to protect this vital vascular bed, as reported in a study by [42]

During insulin resistance, the PI3K/AKT pathway downstream ISR is impaired while the MAPK signaling is functional [43]. As a consequence of this, the compensatory hyperinsulinemia related to the status of insulin resistance induces an overstimulation of the MAPK pathway, resulting in the hyperactivity of the ET-1 system, while the NO bioavailability is impaired and cannot balance the increase in vascular tone [44]. Conditions associated with impaired endothelial vasodilator response to insulin may, therefore, decrease the delivery of nutrients to peripheral tissue, enhancing in turn insulin resistance. There are many evidences on the role of ET-1 in decrease insulin sensitivity in muscle cells [45], and the use of ETA receptor blockers in increasing the glucose utilization by peripheral insulin target cells [46,47].

Other mechanisms that could be involved in vasodilator dysfunction linked with insulin resistance are the reduced endothelium derived hyperpolarization (K channel dysfunction) and altered Rho kinase inhibition in the smooth muscle cells [48,49].

Taken together, these findings indicate that vascular dysfunction in obesity results in an imbalance between NO vasodilator pathway and ET-1 vasoconstrictor activity towards the latter, where ET-1 contribution to the endothelial function is powerful and is associated with impaired in NO- bioavailability.

## 5. Perivascular Adipose Tissue (PVAT) and Fat Accumulation Mechanisms Linked with Endothelial Dysfunction and Insulin Resistance 

The last but not the least in order of relevance is the impact of perivascular adipose tissue (PVAT) on the protection and development of vascular insulin resistance [50]. The adipose tissue is an endocrine organ with immune functions, beyond its well-known role as an energy storage depot.

In general, adipose tissue synthetizes several adipokines that exert different functions on glucose and lipid metabolism, on body weight control, and in organs or tissues insulin sensitivity, including regulation of vascular tone by local autocrine/paracrine actions and systemic endocrine effect [51]. Among those: adiponectin, whose plasma concentration is reduced in obesity and T2DM [52], and inversely correlates with BMI and visceral fat [53,54], enhances insulin- mediated glucose uptake in skeletal muscle, increases insulin sensitivity in the liver, and decreases glucose production by gluconeogenesis [55]. It improves endothelium vasoactivity stimulating NO production via the PI3K pathway [56], and the endothelium redox state under oxidative stress stimuli [57]. Leptin is involved in the regulation of energy intake and appetite and serves as a mediator of the adaptation to fasting [58]. Plasma leptin levels correlate with fat stores and respond to changes in energy balance. The increased circulating leptin in obese patients is related to a kind of systemic leptin resistance [59]. On endothelial cells and smooth muscle cells, leptin acts to stimulate the NO system [60], but long exposure to leptin induces a reduction in NO bioavailability likely increasing oxidative cellular stress [61]. This shows that hyperleptinaemia in obese patients is correlate with increased adverse cardiovascular outcomes, and by itself it is considered an independent risk factor for coronary artery disease and a predictor of acute myocardial infarction [62]. A summary of adipokines functions is listed in Table 1.

The expansion of visceral adipose tissue occurring during obesity correlates with an abnormal synthesis of vasoactive adipokines [63], increased NEFA, and proinflammatory cytokines release into the bloodstream that lead to a systemic low grade inflammation and impair vascular homeostasis towards an increased ET-1 system activity [64,65,66].

It is well known that obese adipose tissue is characterized by adipocytes hyperplasia and hypertrophy and by macrophage infiltration due to preadipocytes and endothelial cells secretion of monocyte chemoattractant protein-1 (MCP-1) in response to cytokines (TNF alfa, and IL6) [67,68,69].

Thus, fat tissue becomes an important source of inflammatory cytokines, which are involved in the development of insulin resistance (by inhibition of insulin dependent glucose uptake) [70], and endothelial dysfunction (impairing vasodilation endothelium NO dependent [71], and endothelium independent, likely the one associated to increased oxidative stress [72].

Recent findings recognize in PVAT an important regulator of vascular tone and functions [73]. Thus, the PVAT results to have vasodilator, anti-contractile, and anti-proliferative functions under physiologic conditions mediated by adipokines, like adiponectin or molecules with vasodilator activity like angiotensin-(1–7) acting in an autocrine or paracrine manner [74,75]. Studies on PVAT human small arteries demonstrated that adipocytes secrete factors (e.g., adiponectin) modulate vasodilation by influencing NO bioavailability. This function of local vascular tone modulator is lost in obesity related metabolic syndrome with evidence of fat hypertrophy, and increased production of inflammatory cytokines (TNF alfa, and IL6) resulting in increased oxidative stress and hypoxia (reversible by use of cytokines blockers, and free radical scavengers) [76,77]. PVAT adipocytes exhibit a different phenotype compared to visceral and subcutaneous depot, with a different reaction to fat intake, since modifying their genes expression and cytokines production towards a proinflammatory state leads to perivascular dysfunction [78].

On the other hand, insulin resistance induced by local inflammation is the link between proinflammatory cytokines and vascular dysfunction mediated by perivascular adipocytes and, which results in impaired NO synthesis and vasoconstrictor tone [79] TNF alfa produced by adipose perivascular tissue is the major mediator involved in this dysfunction [73].

The mechanism by which TNF alfa induces insulin resistance in endothelial cells has been well characterized. It acts through IRS-1 and interferes downstream with the PI3K Akt/eNOS pathway via a p38 MAPK-dependent mechanism. This leads to impairment in the vascular production of NO and enhancement in ERK1/2 and AMPK phosphorylation independently by the p38 MAPK pathway [80,81] (Figure 1). In clinical studies, vascular insulin resistance induced by obesity was ameliorated following the use of TNF alfa blockers, likely in relation to a decrease in oxidative stress [72]. In obese patients, the impairment of vascular insulin signaling is also associated to the excess of FFA in adipose tissue. NEFA interfere with insulin signaling through the same pathways of TNF alfa, downstream the IRS1–2, reducing PI3K activation, and at the end resulting in impaired NO production [82,83]. Taken together, all these findings suggest that adipose tissues around the arteries play an important role in the pathogenesis of vascular dysfunction in the obese-metabolic syndrome condition. In obesity, PVAT loses its dilator and anti-inflammatory actions, leading to a proinflammatory state which promotes vasculature insulin resistance, impairs insulin vasodilation NO dependent, and leads to vascular dysfunction (Figure 1).

Similarly, it has been hypothesized that deposits of fat tissue on the origin of arterioles supplying the skeletal muscle have vasoactive and “vasocrine” role, interfering with post prandial insulin signaling in over feed setting. Insulin induces vasodilation and increases nutritive capillary recruitment in order to allow glucose upload after a meal, through a vasodilator NO dependent action. In overfeeding, as well as in all insulin resistance settings, the PVAT arterioles modulates the insulin signaling through adipocytokines production and secretion in the lumen in paracrine fashion (TNF alfa), inducing vasoconstriction and blood flow redistribution toward adipose tissue, in order to rescue the muscle from the excess of the substrate delivery [73].

In conditions of obesity, DMT2, and insulin resistance, vasoconstriction seems to also involve the adipose tissue, resulting in blunted post prandial adipose tissue perfusion, low blood flow, hypoxia, and increased inflammation. In healthy conditions, the mechanism by which insulin induces adipose tissue vasodilation seems to be through an indirect systemic effect due to increased sympathetic hyperactivity either through the central nervous system [84] (via paraventricular nucleus of the hypothalamus) [85] than through downstream beta adrenergic stimulation [86,87,88]. A down regulation of these adrenergic receptors, due to a sympathetic continuous overstimulation as effect of hyperinsulinemia, may account for the lack of post prandial adipose tissue perfusion in case of insulin resistance [89]. 

These effects result in hypoxia, reduced glucose uptake from target organs, an increase in insulin resistance, and increase ectopic lipid accumulation (liver, muscles) [90].

## 6. Gut Hormones

In the last decade, mounting evidence demonstrated the central role of gut hormones in the regulation of glucose homeostasis, insulin secretion, appetite, body weight, regulation of immune system, and several other physiological functions [91]. Gut hormones are molecules produced by enteroendocrine cells (EECs), specialized secretory cells scattered throughout the mucosal epithelium of the gastrointestinal tract, in response to several stimuli from luminal ingested food [92,93,94,95]. Gut hormones achieve their effects acting on different targets within and outside the gastrointestinal tract. Currently, about 20 different gut hormones have been discovered, some of which exert multiple physiological functions and some with overlapping physiological roles. They recently have attracted more attention because of their role in controlling metabolism in healthy and disease, as implicated in the regulation of insulin secretion and appetite control. Their role has a particular translational interest for future therapeutic approaches on metabolic disorders [96].

### 6.1. Gut Hormones in Metabolic Disease and Obesity

As we discussed previously in this article, obesity and insulin resistance associated to central fat distribution are closely related to an increase in cardiovascular risk and diabetes [26,28].

Achieving glycemic control in diabetes patients is associated with reduction of cardiovascular outcomes [97] as well as weight loss by itself can ameliorate diabetic hyperglycemia [98]. However, the experience obtained through bariatric surgery indicates that beside insulin action, other hormones secreted as consequence of food intake may influence glucose metabolism [99]. Indeed, insulin release after an oral glucose load results higher than after isoglycemic intravenous glucose administration, suggesting a role of gastroenteric tract in stimulating insulin secretion after food ingestion [100].

A number of gut hormones called incretin system, which include Glucagone like peptide 1 (GLP1), Glucose-dependent insulinotropic hormone (GIP), and Oxyntomodulin, are central to this function. They are secreted rapidly a few minutes after a meal by EECs, historically identified as L cells in the distal small intestine (GLP-1, and Oxymodulin), and K cells in the proximal gut (GIP) [101,102].

The half-life of incretins is very short, since they are subjected to a rapid degradation by the dipeptidyl peptidase IV (DPP4) enzyme [103,104].

The incretin hormones stimulate glucose dependent insulin secretion and sensitizing pancreatic beta cells to glucose, which result in a modulation of post prandial glucose excursion [105]. A summary of incretins functions is listed in Table 2.

### 6.2. GLP-1, GIP, and Oxyntomodulin

GLP-1 has generated great interest over time because of its multiple metabolic effects mediated by GLP-1 receptors that are expressed in many different tissues [130]. GLP-1 metabolic actions include the influence on pancreatic beta cells survival and proliferation, pancreatic alfa cells inhibition of glucagon secretion, the reduction of lipid secretion, and glucose production by the liver. On the other hand, GLP-1 inhibits gastric motility resulting in the delay of nutrient absorption; it increases the sense of satiety, reduces food ingestion, and promotes weight loss (Figure 2). GLP-1 exerts other systemic roles directly or indirectly modulating inflammatory response in multiple sites, including heart and blood vessels, reducing oxidative stress and platelet aggregation, improving plaque stability, left ventricular function, and vascular vasodilation [106].

GLP-1 restores insulin stimulated vasodilator reactivity reducing vascular oxidative stress. Clinical studies have shown an improvement of insulin mediated endothelium dependent and independent vasodilation in patients with metabolic syndrome [131].

Recent observations pointed the attention on incretin-based therapy for diabetes in addiction with the traditional care regimen. The use of degradation-resistant GLP-1 analogues (liraglutide in the LEADER trial, and semaglutide in the SUSTAIN-6 trial) in patients with T2DM showed a reduction of cardiovascular mortality, non-fatal acute myocardial infarction (AMI), and stroke compared with the placebo group. All end points achieved by incretins were independent by the little improvement of glycemic control obtained [132,133,134].

The most translation relevance of GLP-1 is due to its insulinotropic effect which results preserved in patients with T2DM, unlike GIP, with significant consequence on developing GLP-1 based therapy to improve insulin resistance.

Another clinical use of this therapy is to enhance weight loss in obese subjects, because of the anorexigenic effect on satiety mediated by GLP-1 receptors located at different sites in the brain [135,136] It is important to make note of the various neural substrates through which GLP-1 and its analogs act in reducing food intake, including the hypothalamic area of the brain (arcuate nucleus of the hypothalamus, periventricular hypothalamus, lateral hypothalamic areas) [137]. These evidence allow us to consider this hormone as a pivotal target for controlling both central and peripheral mechanisms underlying obesity.

GIP is secreted by the proximal small intestine K cells in response to nutrients and acts by binding GIP receptors expressed on pancreatic beta cells, adipocytes, bone, and CNS [138,139,140]. GIP, together with GLP-1, under glycemic stimuli, increase insulin secretion covering 70% of post prandial necessity, promote beta cells proliferation, and suppress beta cell apoptosis [107] (Figure 2). However, since the GIP insulinotropic effect is impaired in T2DM patients, and is reduced in first degree relatives [141,142], while an enhanced glucanotropic effect is seen in the same conditions, less attention has been given to this molecule as a therapeutic target [143]. GIP shows also anabolic proprieties as it promotes adipocytes fat storage and inhibits lipolysis [144], although its obesogenic effect cannot be confirmed in all setting. For instance, mice overexpressing GIP are leaner than wild type [145] because more factors are involved in the control of energy balance, and this process seems to be mediated by GIP modulation of adipokines and other gut hormones secretion.

Oxyntomodulin (OMX) is a peptide that contains glucagon’s amino acid sequence (29 aa) plus an octapeptide tail [108], co-secreted from intestine L cells with GLP-1 in response to nutrient intake. Like GLP-1, OXM is an incretin, which directly induces insulin release from pancreatic islet cells. OXM exerts anorectic effect and increase energy expenditure by different mechanisms that have been incompletely characterized. To date, no specific receptors for OXM have been identified, but it is known that it can activate glucagon (GSIS) and GLP-1 receptors (GLP1R) even whether with less potency than native agonists [109,110]. Most likely OXM plays its anorexigenic effect through GLP-1 R, as confirmed by knockout (K/O) GLP-1R mice studies and pharmacological blockers [146,147,148,149]. These evidences led to development of ‘dual or tri agonist’ analogues (GLP-1R/GCGR/GIP) to obtain anti-obesity effects [150].

### 6.3. Ghrelin and Obestatin

Ghrelin is secreted by the P/D1 closed type cells in the gastric fundus as a prohormone, which requires cleavage and post translational acylation, by the enzyme ghrelin O acyltrasferase (GOAT), to be functional on its own receptors, the growth hormone secretagogue receptor (GHSR 1a) [114]. GHSR 1a is expressed mainly in the central nervous system (CNS) and in the small intestine and pancreatic islets [151]. Ghrelin has multiple effects and exerts its orexigenic action in the hypothalamus by stimulating the Agouti related peptide/neuropeptide Y (AgRP/NPY) neurons and inhibiting the anorectic Proopiomelanocortin/Cocaine and amphetamine-regulated transcript (POMC/CART) neurons in the arcuate nucleus (ARC) [111,112]. Moreover, as it has been demonstrated in pre-clinical studies, the stimulus of AgRP/NPY neurons by different mechanisms, such as circulating levels of prostaglandins, may induce orexigenic and anorexigenic effects with a significant role in central energy metabolism [152,153] 

Ghrelin plasma levels are elevated during fasting and in conditions related to malnutrition like cachexia and anorexia nervosa [154]. By contrast plasmatic Ghrelin is reduced in obesity and in insulin resistance, T2DM, and hypertension [155]. Ghrelin secretion is controlled by cholinergic stimuli [156] and even if in vitro, is stimulated by noradrenaline and can be suppressed by a b1 blocker, atenolol [157].

Ghrelin is considered a key regulator of glycemic homeostasis by inducing the enhancement of GLP-1 secretory response to nutrients. Since plasmatic Ghrelin peaks during fasting, it acts by preparing the target cells to release GLP-1 following food ingestion. Evidence in animal models showed an enhanced glucose-stimulated GLP-1 release after Ghrelin treatment, and an improvement of glucose tolerance, which are blunted in presence of GLP-1 receptor blockers or in case of GLP-1R KO mice [158].

In addition, Ghrelin is involved in the modulation of lipid metabolism and body composition [159], and it has been linked to anti-inflammatory [160,161] and antiapoptotic effects [162].

Ghrelin exerts relevant effects on cardiovascular systems as GHS receptors are found in the heart, where they induce positive inotropic effects, and in the vasculature, where it leads to an improvement of blood pressure by lowering peripheral resistance [163]. Clinical studies showed that intra-arterial Ghrelin administration in obese patients with metabolic syndrome restores endothelial NO dependent vasodilation [113].

Ghrelin acts not only by enhancing NO bioavailability in endothelial cells but also by reducing the ET-1 imbalance vasoconstrictor action in obese patients. Exogenous Ghrelin blunts the vasodilator action of ET-1 blocker BQ123 in obese but not in lean subjects and increases vasoconstriction reactivity to the LMNA (NG-monomethyl-L-arginine), and to the NO synthase inhibitor. These evidence confirm that Ghrelin can normalize the balance between vasoconstrictor (ET-1) and vasodilatation (NO) mediators by increasing NO production [24]. Same results were obtained by using Ghrelin in hypertensive patients modulating oxidative stress [164].

Obestatin results from an alternative splicing of the Ghrelin gene, as a 23 amino-acid peptide derived from the common precursor prepro-ghrelin [115].

The precise identity of the cognate receptor for obestatin has still not been determined. Some findings suggest that obestatin may signal thought the GLP-1 receptor, since the same effect on some target cells, but current knowledges are insufficient to draw any conclusion. Obestatin’s half-life is very short because it is degraded by enzymatic proteases. Obestatin has anorectic effects and can modulate metabolic homeostasis, improving insulin sensitivity and glycemic control by glucose uptake, promoting pancreatic beta cell proliferation and survival, inhibiting lipolysis in adipose tissue. Obestatin also appears to be involved in blood pressure regulation, promotes cardioprotective actions [116], and exerts beneficial effects on endothelial function by increasing NO production [117].

The intra-arterial infusion of increasing doses of obestatin results in augmented vasodilation in obese and lean subjects; this effect is blunted following infusion of L-NMMA, an NO synthase inhibitor, confirming that obestatin acts by enhancing NO-dependent vasodilation in the human circulation. This effect is preserved in obese patients, where it is accompanied by reduced ET-1-mediated vasoconstriction as shown by the use of BQ 123, ET-1 blocker whose effect in obesity does not enhanced by obestatin [118].

### 6.4. PYY and Insulin like Peptide 5

Peptide tyrosine tyrosine (PYY) is secreted after meal stimulation by L enteroendocrine cells together with GLP-1 [119], along the proximal and distal intestinal tract. Whereas GLP-1 is more present in the proximal tract, PYY is produced in a large amount in the distal ileum toward the colon [165], and its release is triggered under neurohormonal control [166], and responses to metabolites of the gut microbiota [167] [168]. The same gene encodes for two proteins, PYY 1–36 and PYY 3–36, resulting from DPP4 cleavage of the entire form [169]. Both have a considerable role in the “ileal brake” function (together with cholecystokinin, GLP1), a local feedback signal which inhibits gastric emptying, pancreatic secretion, and intestinal mobility induced by food intake [120,121] in order to slow down the transit and increase the nutrients absorption in the upper gut. PYY has anorectic effect [122], acting through the Neuropeptide Y receptors (NPYs) located on enterocytes, myenteric, and submucosal neurons, afferent fiber, and in CNS [166] (Figure 2). PYY in high doses induces nausea and anorexia [170], which is why this peptide has not been considered a good therapeutic target for obesity. In fasting humans PYY is inversely related to BMI. Because of the distal location of enteroendocrine cells secreting PYY, a high amount of this hormone is produced in case of bariatric gastric surgery (tenfold higher), when increased flow of nutrients is delivered to the distal gut and directly stimulates L cells to release the hormone [171,172]

In physiological conditions, PYY is secreted likely by paracrine and neural mechanisms. PYY does not have clear metabolic effect to date, and PYY infusion in human does not modify glycemic control, insulin, or glucagon levels [173,174].

Insulin like Peptide 5 (INSL-5) is co-secreted with GLP-1 and PYY by the colonic L enteroendocrine cells where they are co-stored into separate vesicular, [123], and represents the major enterohormone produced from large intestine. In contrast with the GLP-1 and PPY function, INSL-5 is considered an orexigenic hormone; indeed, intraperitoneal administration increase food intake in mice, acting through the Relaxin/Insulin-like family peptide receptor 4 (RXFP4) [124], behavior lost in K/O RXFP4 mice [175]. The INSL-5 secretion by Colonic enteroendocrine L cells most likely is modulated by several factors. Fasting or restricted caloric diet increases INSL-5 colonic and plasma levels and are normalized after food intake. On the other hand, germ free mice have shown an increased INSL-5 expression, and a high concentration in the lumen of short chain fatty acid, bile acids, produced by intestinal bacteria reduces INLS-5 secretion. These last evidence suggest the role of some microbial metabolites in signaling for gut hormones secretion and INSL5 as a link between metabolism, microbiota and host [176]. Other systemic factors such as angiotensin II (Ang II) and arginine vasopressin (AVP) could modulate INSL-5 secretion [123].

The effect of INSL-5 on metabolic homeostasis is less clear, and the role of INSL5 in glucose control and in insulin sensitivity is still debated.

### 6.5. Cholecystokinin

Cholecystokinin (CKK) is one of the first gut peptides discovered with the ability to regulate the digestion through stimulation of gallbladder contraction. Today many other functions have been assigned to CCK including antinflammatory action [177], the modulation of cardiovascular functions [178], the inhibition of gastric acid secretion [125], insulin secretion [126], and the sense of satiety [127]. 

CCK is secreted by the enteroendocrine cell historically known as I-cells, located closely to the K-cells GIP releasing in the duodenum and proximal jejunum, and close to L-cells GLP-1 releasing in the distal part of the enteric [179]. As previously reported, it is not surprising that K-cells secret CCK [95], and I-cells can release GIP or other gut hormones [180]. CCK is secreted after meal in response of fat and protein nutrients [128], and plasma level are enhanced of 10–20 folds. Most likely, CCK secretion from the EECs is triggered by contact of ingested food with the secretory cells in the duodenum. The immediate effect is on gastric empty control but also on signal by the vagal afferents to the central brain in order to inhibit appetite.

The encoded form of the CCK gene is a 115 amino acid polypeptides, which eventually will be differentially cleaved in many truncated circulation forms known as CCK-8, CCK-33, CCK-58 peptides [181]. CCK acts through CCK1 (located in peripheral tissue), and CCK2 receptors (located centrally). Most of its actions occur by the CCK1 receptors, which are also found in visceral afferent fiber vagal. When CCK1 receptors are stimulated they may signal satiety to hypothalamus [129].

Beside the regulation of energy control, CCK may also exert incretin effects, thereby enhancing insulin secretion by beta pancreatic cells [126], promote beta cells survive, and suppress apoptosis [182,183]. CCK represents an interesting target for the development of obesity-diabetes therapy, as shown by in vitro studies and animal models based on the administration of CCK peptide analogues [184].

## 7. Obesity and Gut Microbiota

The gastrointestinal tract is colonized by the gut microbiota which consist of more than 100 trillion microbial cells, spread based on a concentration gradient along the gastrointestinal tract, with the highest density in the colon [185].

The gut microbiota collaborates with the host to accomplish many physiological activities including the improvement of enteric digestion and absorption of nutrients, the defense against colonization by pathogen microbes, the maintenance of the gut barrier, the modulation of inflammation and of the entero-hormones secretion by the ECCs, the promotion of immune system maturation, and the influence on the formation of capillary networks in the small intestinal mucosa [186,187,188,189].

In physiologic conditions, host and microbiome cohabit in symbiosis, so that the microbiome contributes to health maintenance and the host provides a favorable environment for its survival. Mounting evidence shows that quantitative and qualitative alterations of gut microbiota composition, called “dysbiosis”, may represent a new risk factor for developing future diseases, including obesity. 

It has been shown that the intestinal microbiome can be responsible for the efficiency of expenditure or harvest energy from food intake and has a role in the development of obesity by altering host energy storage and harvest [190,191,192]. The microbiome also seems to affect the susceptibility to insulin resistance and promotes the development of non-alcoholic fatty liver disease [193].

Despite intraindividual variability, 5 phyla are prevailing in the intestinal community: Actinobacteria, Bacteroidetes, Firmicutes, Proteobacteria, and Verrucomicrobia. Environmental factors govern the distribution along the gastrointestinal tract, especially the pH and the oxygen content (mainly in the stomach and duodenum). Thus, the proximal trait is enriched of species that can survive at low pH and high oxygen like Firmicutes (Lactobacillaceae) and Proteobacteria (Enterobacteriaceae), while the large intestine presents an increasing of diversity with several obligate anaerobic bacteria as Bacteroidetes (Bacteroidaceae, Prevotellaceae, Rikenellaceae), and Firmicutes (Lachnospiraceae, Ruminococcaceae), and Akkermansia muciniphila, (as representative of Verrucomicrobia) [194].

Most of the functional consequences of alterations of the gut microbiota into changes in health and susceptibility to disease derived from pre-clinical animal studies [195]. For instance, mice have been very helpful to evaluate mechanisms influencing microbiota composition, and there is significant interest in mouse gastrointestinal microbiota as an easily manipulable model for research studies. Moreover, animal models were of great interest to investigate the potential of fecal microbiota transplantation (FMT), which has been proposed as a therapeutic approach to directly introduce intestinal flora. The FMT in mice has been already demonstrated to increased proportion of *Firmicutes/Bacteroidetes* and then to reduce the level of vascular inflammation and cardiovascular disease in this animal [196].

In obese subjects, the bacterial diversity in large intestine is decreased, with difference in microbial composition [197], resulting in microbial enzymatic alterations [198]. Obesity is associated with a reduction in Firmicutes/Bacteroidetes ratio [186], which increases after weight loss following dietary restrictions and the Roux-en-Y gastric bypass intervention [199,200]. Unfortunately the reproducibility of these studies in humans is limited by differences in race-ethnicities, diet, and metabolic diseases, and other investigations have found conflicting findings [201].

On the other hand, it is known that gut microbiota is influenced by an interplay of genetic, immunologic innate, and environmental factors. Among them, diet is a major factor driving the composition and the metabolism of gut microbiota and has been associated more than the others to obesity [202,203]. Indeed, the dietary intake of processed foods, beverages with sugar, and red meat has been correlated to obesity [204] as well as to altered microbiota composition. In contrast, intake of vegetables, fibers, and yogurt are associated to weight loos and to a different fecal microbial composition [187,205].

## 8. Gut Microbiota, Energy Harvest, and Storage

The mechanisms by which gut microbiota may drive obesity have not been elucidated yet. However, there is evidence supporting a correlation between microbial active metabolites with energy harvest. Bacteria fermentation of food can influence the nutritional intake of the host up to 10% of energy. Fiber digestion through bacterial fermentation in the colon leads to the production of short chain fatty acids (SCFA) as acetate, propionate and butyrate. Studies from germ free models have demonstrated that SCFA are almost undetectable in the absence of gut microbiome [206]. Interestingly, germ free mice appear to be resistant to diet-induced obesity (High Fat Diet) and to have low levels of inflammatory cytokines (TNF alfa) and improved insulin sensitivity [207]. Butyrate promotes gut barrier by reducing luminal pH. However, it is also a source of energy for colonic cells and for bacteria [208]. The importance of this pathway in human increases in patients with sugar malabsorption when delivery of unabsorbed sugar is increased to the colonic lumen. Acetate and propionate are absorbed rapidly and used as energy source for hepatic cells. Further SCFA released in small amount in circulation, participate to different metabolic effects and brain related host-signaling mechanism [209]. In animal models, SCFA have been shown to act on liver and muscle cells through the energy sensor AMP-activated protein kinase stimulating glucose uptake and fatty acids oxidation with a regulation of gluconeogenesis, lipogenesis and energy metabolism [210]. Moreover, propionate can directly initiate a gut-brain neural circuit acting as agonist of FFAR3 (Free Fatty Acid Receptor 3) in the periportal afferent neural system to induce intestinal gluconeogenesis with beneficial effects on host physiology [211].

In addition, gut microbiota influences energy homeostasis and energy harvest through bile acids metabolism. Intestinal bacteria can modify bile acids through a deconjugation process. Furthermore, some specific strains from the clostridium genus can also dehydroxylate the unconjugated molecules in the colon. It is known that bile acids are endogenous ligands for the nuclear receptor, Farnesoid X receptor (FXR) [212] and for membrane G-protein bile acid receptor (TGR5) [213]. FXR is a ubiquitously expressed nuclear receptor found in different tissues, such as liver, intestine, kidney, adipose tissues and immune cells. FXR-mediated bile acid signaling plays a role in the maintenance of lipid and glucose homeostasis, as shown by the evidence in FXR-null mice of impaired insulin signaling with dysregulated glucose homeostasis [214,215] and elevated blood cholesterol and triglyceride levels. The G protein coupled bile acid receptor 1 (TGR5) is a member of GPCR membrane proteins and is ubiquitously expressed in diverse tissues, including muscle, adipose tissue, immune cells, endocrine organs, and the intestinal tract [216] with a known role in thermogenesis, energy expenditure and energy homeostasis [214]. TGR5- deficient mice show severe metabolic syndromes including obesity, insulin resistance and impaired glucose and lipid homeostasis [217]. TGR5 has been reported to promote incretin secretion like GLP-1 from the intestinal L enteroendocrine cells, therefore bile acids secreted after meal control the release of insulinotropic hormones [218]. Gut microbiota-dehydroxylated bile acids are more hydrophobic, with more efficacy to bind both nuclear receptors amplifying the signals downstream.

Bile acid signaling is an important mediator of the beneficial effects of sleeve gastrectomy surgery in rodent models [219]. Fecal transplant from mice who have undergone to Roux-en-Y gastric bypass (RYGB) surgery to germ free mice leads to a significant weight loss and decreased fat mass compared with sham controls, supporting the role of gut microbial composition in weight loss and the reduction of adiposity after RYGB surgery [220]. These evidences in animals, could explain the beneficial effects in human of the RYGB surgery, whose metabolic improvements and mortality reduction rates are not justified by only weight loss and by lowering caloric intake [221]. 

The gut microbiota not only increases energy uptake from food intake, processing undigested molecules and modifying endogenous metabolites, but also promotes the harvest of calories in adipose tissue by signaling involved in energy storage. Indeed, the lipoprotein lipase inhibitor Angiopoietin-like 4 (Angpl-4) is downregulated by the gut microbiota, resulting in increased LPL activity and in enhanced hydrolysis of triglycerides with uptake of fatty acid in adipose tissue. In GF mice higher level of Angpl4 and lower LPL activity with less energy storage [190] is present. In humans, levels of Angpl-4 are higher in twins with lower BMI as compared to their obese counterpart [222]. 

Another signaling influenced by gut microbiota is the AMP-activated protein kinase (AMPK) sensor [223], which regulates energy producing pathways. Evidences show that SCFA bacterial production is a possible direct AMPK activator [224]. In mice, AMPK is activated by both venous infusion and oral administration of SCFAs, explaining the regulatory effect of gut microbiota on AMPK activity [225]. The activation of AMPK develops the phosphorylation of acetyl-CoA carboxylase, and the reduction of malonyl-CoA inducing utilization of fatty acid by its oxidation, as energy source [226].

## 9. Gut Microbiota and Satiety Signaling

Gut bacteria may have a specific role in the regulation of host appetite. Indeed the symbiotic interactions between intestinal microbiota and their host influence brain functions and behavior [227].

Different metabolic phenotypes are associated with similar gut microbial composition (in obese there is a reduction of relative proportion of Bacteroidetes/ Firmicutes) [186] and transplant of microbiota from obese phenotype to germ free mice results in to increase total body fat, suggesting a role of microbiota in energy harvest [191].

Following those evidences and in line with the recent findings that gut bacterial proteins expressed in their different growth times can modify the host appetite control [228], it is possible to hypothesize a bacterial-host homeostatic model of appetite control that integrates gut bacterial growth dynamics and host molecular pathways controlling energy homeostasis [229].

Gut bacterial growth is constantly influenced by nutrient supply [230] and by the host chemical (digestive gastric and bile acid juices, digestive enzymes), and physical (intestinal peristalsis, colon contractions preparatory to bacteria elimination by defecation) factors.

Regular daily meals trigger bacterial growth (from exponential growth to stationary phase in 20’ after nutrients supply), and according with the different growth phases induced by nutrient supply (exponential or stationary phases), gut bacteria express different molecules [231], which may act locally in a paracrine way through intestinal mucosa, or in an endocrine/systemic way to the brain, involving the short term and long term regulation of feeding behavior.

Studies carried out on E. Coli, the most abundant facultative anaerobe gastrointestinal bacteria, have shown as E. Coli proteome profile in stationary (Stat) growing phase is more abundant and is different than the one in exponential phase (Exp). Between all molecules, E. Coli expresses and secretes the caseinolytic peptidase B (ClpB), a bacterial protein mimetic of alfa MSH that can activate POMC/ARC neurons, signaling anorexigenic message (which induce satiety in the host). Intraperitoneal injection of Stat E. Coli proteins activates anorexigenic neurons in the brain [228]. The role of ClpB protein in the physiological and pathological regulation of eating behavior is shown by its higher level in the plasma of patients with eating disorders than healthy subjects. 

The intestinal infusion of E. Coli proteins from different growth phases correlate with plasma increased GLP1 (exp) and PYY (Stat) levels. [232,233]. Of note, the bacterial protein release, in Exp bacterial growth phase stimulates GLP-1 secretion improving glucose metabolism, and similarly the one released in Stat phase can stimulate PYY, 20’ min after started meal, developing satiety signal. Those mediators produced by intestinal enterocytes under influences of gut microbiota and the ClpB, expressed by the bacteria, are responsible of short and long term appetite control respectively [229]. 

All this evidence supports an integrative model in which the appetite regulatory centers are affected by host signals, and bacteria derived signals, in the different phases of growth after nutrient supply and in response to the homeostatic needs. 

## 10. Gut Microbiota, Insulin Resistance, Vascular Disfunction, and Cardiovascular Disease

Microbial metabolites, which results as product from diet nutrients and different kind of microbial population, can interfere with host metabolic functions and have a role in pathophysiology of metabolic disease. SCFA, indole, bile acids, and LPS are among the most important microbial products with known bioactivity in stimulating EECs secretion of gut hormones [234] (Figure 2). Moreover, it is recently reported that aromatic amino acid metabolism mediated by microbiota is associated to toxin production, involved in endothelial disfunction and cardiovascular disease [235]. Acetate, propionate, and butyrate are SCFA, resulting from the anaerobic fermentation of undigested dietary starch, fiber and other polysaccharides reached the colon, which can be absorbed by the colonic mucosa and have systemic effects. SCFA provide from 5 to 10% of host energy intake, but their role is much broader and they seem to have a function on inflammatory responses, modulation of autonomic system and interfere on many cellular functions included chemotaxis, phagocytosis, ROS stimulation, cell proliferation, histone deacetylases inhibition, and intestinal barrier integrity [236]. It has been identified some G protein coupled receptors as target of SCFA, spread on intestinal epithelial cells, adipocytes, immune cells, smooth muscle cells of small vessels (for tensive control), renal juxtaglomerular apparatus (for renin secretion), that justify the broad spectrum of effects associated with SCFA [237,238,239]. Not only products of bacterial metabolism or proteins secreted by them, but also molecules produced by the breakdown of the bacterial wall play a role in the modulation of host metabolism: polysaccharides, peptidoglycans, and lipoproteins are bacterial derived molecules (from intestinal gram-negative bacteria) that can activate the Toll Like receptors (TLR) expressed by enteroendocrine cells on gut epithelium [240].

TLRs are specialized pattern recognition receptors (PRR) which stimulate host immune response when activated by pathogen-associated microbial products (PAMPs) like LPS [241]. Lipopolysaccharide (LPS) and peptidoglycans might act on gut epithelial cells, but they can also translocate through the intestine and reach several target tissues triggering pro-inflammatory response. It has been shown that mice under high feed diet (HFD) underwent induced obesity (DIO) and insulin resistance associated with finding of plasma high levels of LPS, TNF alfa, IL1 and IL6, which are all proinflammatory markers [242]. This evidence links the low grade inflammation found in obesity and diabetes related insulin resistance with gut microbiota and diet [243]. Indeed, many studies have confirmed the association between high levels of LPS or proinflammatory markers and inhibition of insulin pathways on target organs, justifying the found insulin resistance [244,245]. Tlr4 knock out mice show low expression of inflammatory cytokines and do not develop insulin resistance [246]. Primum movens of this mechanism is the LPS/bacterial translocation from the intestinal lumen to other tissues locations where they can induce inflammation. Translocation must occur before the beginning of metabolic disorders and explains their manifestation [246]. Bacterial translocation might occur for enterocytes internalization by phagocytosis [247], or through the innate immune cell phagocytosis and mesenteric lymph nodes dissemination [248]. Therefore, all conditions which impair the gut wall barrier function facilitate the transition, and it is known that HFD mice and obese mice have an increased gut permeability and higher metabolic endotoxemia [249]>.

Gut microbiota can also affect the host through bioactive metabolites that may contribute to development of diseases. A very important observation comes from metabolomics studies which have identified in the trimethylamine N-oxide (TMAO) a gut microbe-derived metabolite, a strong predictor of plaque atherosclerosis and coronary artery disease [250,251,252]. TMAO is an oxidative hepatic product from trimethylamine (TMA), a gas generated by intestinal microbes from the choline, phosphatidylcholine, and L-carnitine, in the colon tract. Red meat, eggs, milk, liver, shellfish, and fish are major sources of lipid phosphatidylcholine in human plasma. TMAO increases 4–8 h after a rich meal and is normalized in 24 h in preserved renal function. TMAO acts on platelets and further enhances susceptibility to thrombosis risk [253]. Mice fed with carnitine or L choline rich diet have shown an increased atherosclerotic plaques development together with higher plasma levels of TMAO, macrophage cholesterol accumulation and foam cell formation. According with the proatherogenic contribution of the TMA/TMAO gut microbiota generation, studies from germ free mice have shown the suppression of TMAO production even under specific diet while the use of a broad-spectrum antibiotic for short term has shown suppression of diet related atherosclerotic plaque development [252]. However, the TMA/TMAO pathway is just one of the many known and not yet known microbiome-dependent pathways that could be involved to diseases pathogenesis. 

In the last decade, gut microbiota metabolism of amino acids and nitrogen metabolites gained a role in endothelial impairment and increased development of cardiovascular disease [254]. The aromatic amino acids in proteins can be metabolized by the gut microbiota [255,256], and host liver [257] to toxins such as indoxyl sulfate, indoxyl glucuronide, indoleacetic acid, p-cresyl sulfate, p-cresyl, phenyl sulfate, and others [258]. 

Further some bacterial products are involved in the communication between the gut nervous system and the central nervous system through sympathetic activation and involvement of immune system [259,260].

About heart failure, many evidences support the gut hypothesis of heart failure that implies together with systemic congestion, intestinal mucosa ischemia and wall edema which promotes intestinal permeability, bacterial translocation, increased endotoxemia and systemic inflammation [261,262] [263,264]. However, even if the association with vascular compliance and function is complex, at the best of our knowledge we did not find any direct association between gut microbiota and endothelial factors other than NO that may regulate these mechanisms. 

All these and other findings demonstrated the important role of the microbiota to promote health and diseases and even if until now the major works have been focused on discover the community of bacteria that are associated with enhanced disease susceptibilities, now it seems to be more relevant the interaction between host diet and microbiota generated metabolites biologically active in order to test novel therapy targeting microbiome or the enzymatic pathway triggered by it.

## 11. The Autonomic Nervous System (ANS)

In the complex mechanisms that regulate food intake, the autonomic nervous system (ANS) has a major role because is involved in appetite/satiety signal and energy storage/expenditure. ANS plays a role through the short-term regulation of body weight, which is affected by the local gut environment. On the other hand, the long-term regulation developed by ANS depends by the CNS response to endocrine signal as resulting of the nutrients absorption. Through long term and short-term regulation, ANS allows the communication between CNS and gastrointestinal system in both ways.

The short term regulation of body weight is mediated by afferent sensorial nerves which control the sense of satiety, by gastric distension and gut hormones release (like GLP1-GIP, CKK) [265]. The afferent sensorial nerves are activated by gastric mechanoreceptors triggered by wall distension and specific nutrients like SCFA and others stimulating gut hormones release [266]. All these afferent signals reach the solitary tract/area postrema complex in the brain, where are elaborated with other peripheral message. After this afference integration process, it generates a new signal that reach the gut via vagovagal autonomic reflexes and control the gastroenteric secretory, motility, and absorption functions [267,268].

The ANS long term regulation of body weight involves the homeostatic system of energy expenditure and storage. ANS acts on energy expenditure by activation of sympathetic nervous system, responsible for fat mobilization from the adipose tissue, and for thermogenesis from the brown adipose tissue [266]. Sympathetic hyperactivity has been found in obese subjects; it is not generalized but is selective to specific systems like the muscle vasculature and kidneys [269,270,271]. Sympathetic hyperactivity in obesity has negative cardiovascular effect including develop of obesity related hypertension and it does not have any favorable effect in enhancing energy expenditure and weigh loss. Therefore, the increased SNS activity might be an ineffective adaptive mechanism to weight gain, according with initial studies [272].

The assumed mechanisms responsible for the sympathetic hyperactivity in obesity related hypertension involve OSAS, impaired baroreflex, increased leptin with central mechanism, and hyperinsulinemia and insulin resistance, among peripheral mechanisms of sympathetic activation.

In conclusion the afferent vagal signals seem to be the major link between gut and the CNS in controlling food intake and body weight regulation. 

Recent discoveries brought additional light to this concept, indeed among the ECCs have been identified a type of sensory epithelial cells that synapses directly with vagal neurons. As kind of neuropod cells, they form a neuroepithelial circuit which connect the gut lumen to the brainstem, transducing sensory stimuli from nutrients (sugar) in milliseconds, using glutamate as neurotransmitter [273].

## 12. The Brain-Gut-Microbiome Axis (BGM)

The term “Brain-Gut-Microbiome Axis” refers to the established bidirectional communication and interaction that involves nervous system, the gastrointestinal organ and gut microbiome [274]. Perturbations in the communication between the parts or alterations at any level, seem to have a role in the pathogenesis of some neurological (Parkinson’s Disease, multiple sclerosis), gastrointestinal and metabolic disorder (IBD, obesity and food addiction) or socio-affective behavior (autism spectrum disorders, depression and anxiety).

The important role of microbiota on healthy development and maintenance of the central nervous system is confirmed by preclinical evidence, involving germ free animal models, broad spectrum antibiotic short-term use, fecal microbial transplantation, colonization of human or synthetic microbiota and probiotic therapy administration [275]. 

However, the clinical studies performed, mostly based on the identification of central nervous system effects by probiotic or prebiotics treatment, need to be expanded. The new technologies are a good opportunity to determine the impact of all microbial community (transient and resident) on diseases [275].

Gut microbiotas send signals to brain through the neuroendocrine and neuroimmune circuits and by the vagus nerve [276] The communication occurs through microbial molecules or their derived metabolites which stimulate enterochromaffin, enteroendocrine or immune systems target cells. SCFA, secondary bile acids (2BAs) tryptophan (Trp) metabolites are the most used messengers in this communication [277,278,279,280] (Figure 3).

The major signaling molecules are SCFA. They are important for the host as energy resource, the regulation of water and electrolytes absorption, and mucosal proliferation. Further they activate target receptors (GPR41 and GPR43) on EEC (L cells) to stimulate GLP1 and PYY secretion which regulate metabolic functions and induce satiety [277,278] (Figure 2).

Bile acid are products of cholesterol metabolism in the liver, after undergoing a secondary metabolism by the gut microbiota [281]. Secondary bile acids interact through ileal nuclear receptor Farnesoid X Factor (FXR) and TGR5 leading to GLP1 release and regulation of glucose homeostasis [218]. In add, secondary bile acids stimulate fibroblast grow factor (FGF-19) production in small intestine. The FGF-19, crossing the blood–brain barrier (BBB), sends signals to the arcuate nucleus for regulation of energy intake [282]. Evidence shows as in mice, intracerebroventricular injection of FGF-19 directly into the brain increases metabolic rate and decreases circulating glucose and insulin concentrations [283,284,285,286] (Figure 3).

SCFAs and 2 BA act on enterochromaffin cell stimulating the synthesis and release of serotonin (5HT) in the lumen. 5HT is important for regulation GI motility and secretion. 5HT is produced by the EEC and stored there and in enteric neurons for the 95% of all body. It derives from tryptophan (Trp), an essential amino acid available only through the diet, and made accessible by gut microbiota [280]. According with the amount of Trp in the diet, bacteria produce theirs signal molecules which stimulate EEC to produce 5HT. ANS modulates the 5HT secretion on the gut lumen directly acting on EEC, and EEC signals to afferent fiber through connections similar to synapsis and neuropods [287].

The bacterial structural molecules (LPS and peptidoglycan) can activate enteric nervous cell signaling locally in the gut (through TLRs family) [288] or systemically. Microbiota structural molecules and metabolites can translocate through the gut barrier and reach the central nervous system by crossing the blood–brain barrier (BBB) [289], interacting with FXR, TGR5, GPR41 expressed in different location of central nervous system [290,291,292]. Gut microbiota can also produce itself neuroactive molecules like gamma aminobutyric acid, 5HT, norepinephrine and dopamine, and stimulate sensitive targets [293,294,295,296].

Noteworthy, SCFAs are indispensable for the correct development of blood-brain barrier [297]: GF mice show an increased in BBB permeability since intrauterine life due to a reduced expression of some proteins belonging to the tight junctions. This evidence confirm how important is the interaction between microbiota and host for organ health and the development of the systems. 

The brain signals to gut microbiota first modulating the gut environment through the ANS, which is responsible for the gut microbiota habitat, the community structure, composition, and activity. Indeed, the ANS regulates the GI mobility and establishes the intestinal transit times. This influences water absorption, nutrient availability, and microbial clearance rates [298,299], promotes the intestinal mucus layer integrity, and contributes to gut barrier healthy by maintenance of permeability (Figure 3). Stressing stimuli are associated with increased intestinal permeability (and enhanced inflammation induced by bacterial molecules) [300,301] while hyperactivity of sympathetic signal (catecholamines mediated) may affect the goblet cells, impairing the quality and quantity mucus production [302,303]. Moreover the ANS influences microbiota, modulating the secretion of gastrointestinal juices (which modifies the pH: gastric juice and bicarbonate), and interferes with antimicrobial peptides production and mucosal immune response [304].

Brain can signal directly to the gut through activation of ECC, neurons, immune cells, which release 5 HT, catecholamines, dynorphin and cytokines [305,306]. These molecules recognize microbial receptors and modulate bacterial behavior sometime increasing virulence [307,308,309,310,311].

In summary, we can affirm that microbiota act as an organ to be considered an indispensable integral part for human health and wellbeing. The microbiota depends on the regular functioning of the intestine (especially in relation to nutrients supply), and itself collaborates in intestinal maintenance. A healthy microbiota affects the wellbeing of the whole organism and the greatest confirmation is the constant double direction dialogue with the brain. Brain function is modulated by the messages sent through directed mediators by the microbiota, and the microbiota in turn depends by CNS activity, for most of its life cycle. Improving the health of the microbiota, with a healthy diet, reducing stress-inducing stimuli, and hopefully by intervening on the microbial flora through pharmacological treatments, could be therapeutic equivalents for diseases associated to dysbiosis and altered colonization. Further studies must be developed to better understand how intervene on microbiota in order to cure related diseases.

## 13. Conclusions

Obesity is one of the major public health challenges of the current century, which has reached the proportions of a global pandemic. Obesity is the main risk factor for developing type 2 diabetes, cardiovascular disease, and specific types of cancers.

The pathogenesis of obesity is complex and implies many pathophysiological mechanisms involved in the control systems of energy harvest and storage, satiety and appetite, the homeostasis of nutrient metabolism, and regulatory hormones and microbiota. This last one acts as an organ and contributes to gastrointestinal functions by intervening in a crosstalk between the intestine and the brain through many types of cells (ECC, EEC, immune system, neurons) and many mediators (SCFA, 2 BA, LPS, 5HT, catecholamine, cytokines). This crosstalk is further exacerbated by the evidence that the negative modulation of vascular function, and then the predisposition to cardiovascular diseases that is characteristic of the obese subjects, is mediated by both gut hormones effect and adipose tissue accumulation and their second products. In fact, as described, excessive adipose deposition increases adipokine release leading to increases in inflammation and recruitment of immune cells [312].

Understanding all these complex mechanisms and intervening in their dysfunctions could represent a novel strategy to reduce this major menace of the 21st century.

## Figures and Tables

**Figure 1 nutrients-13-00613-f001:**
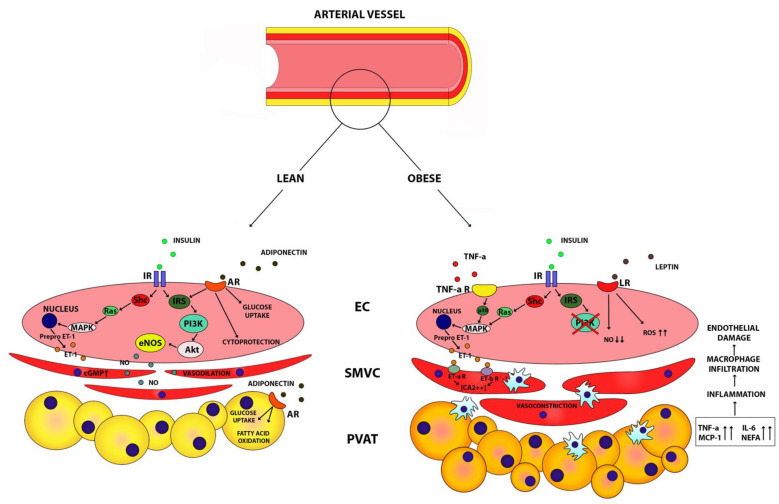
Interplay mechanisms of action between vascular endothelial cells and perivascular adipose tissue in lean and obese subjects. In lean subjects, insulin and adiponectin stimulate endothelial cells to secrete nitric oxide and endothelin 1. This mechanism induces homeostasis between vasodilation and vasoconstriction, promotes glucose uptake and cytoprotection, and the oxidation of fatty acids in adipose tissue. Instead, in obese subjects, insulin resistance, the increased release of leptin and TNF-alpha stimulate vasoconstriction and reduction of NO bioavailability. Insulin resistance, Leptin, and TNF-alpha induce ROS production, the release of pro-inflammatory cytokines with inflammation, macrophage infiltration of PVAT and endothelial damage. Abbreviations: IR, insulin receptor; IRS, insulin receptor substrate; PI3K, phosphoinositide 3-kinase; Akt, protein kinase B; MAPK, mitogen-activated protein kinase; cGMP, cyclic guanosine monophosphate; AR, adiponectin receptor; LR, leptin receptor; TNF-a R, tumor necrosis factor-alpha receptor; NO, nitric oxide; eNOS, endothelial nitric oxide synthase; ET-1, endothelin-1; Prepro ET-1, Prepro endothelin-1; ET-a R, endothelin 1-a receptor; ET-b R, endothelin 1-b receptor; ROS, radical oxygen species; EC, endothelial cell; SMVC, smooth muscle vascular cell; PVAT, perivascular adipose tissue; TNF-a, tumor necrosis factor-alpha; IL-6, interleukin-6; MCP-1, monocyte chemoattractant protein-1; NEFA, non-esterified fatty acids.

**Figure 2 nutrients-13-00613-f002:**
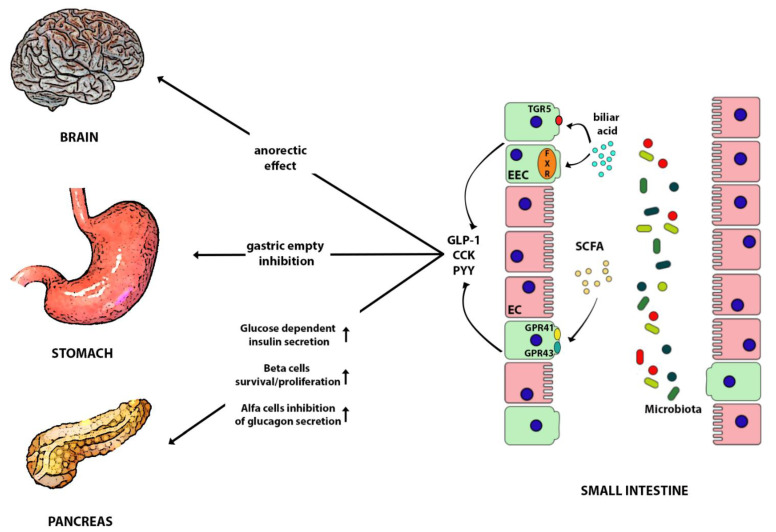
Effect of Gut microbiota in regulating the incretin secretion and their anorectic effects in peripheral organs. Gut microbiotas release chemical mediators, including SCFA which, together with bile acids, stimulate enterochromaffin cells to secrete GLP-1, CCK, and PYY. GLP-1, CCK and PYY induce an anorectic effect, gastric empty inhibition, glucose dependent insulin secretion, beta cells survival/proliferation, and alfa cells inhibition of glucagon secretion. Abbreviations: SCFA, short chain fatty acid; GLP-1, glucagon-like peptide 1; CCK, cholecystokinin; PYY, peptide YY; EEC, enterochromaffin cell; EC, enteric cell; GPR 41, G protein-coupled receptor 41; GPR 43, G protein-coupled receptor 43; FXR, farnesoid x receptor; TGR5, G-protein bile acid receptor.

**Figure 3 nutrients-13-00613-f003:**
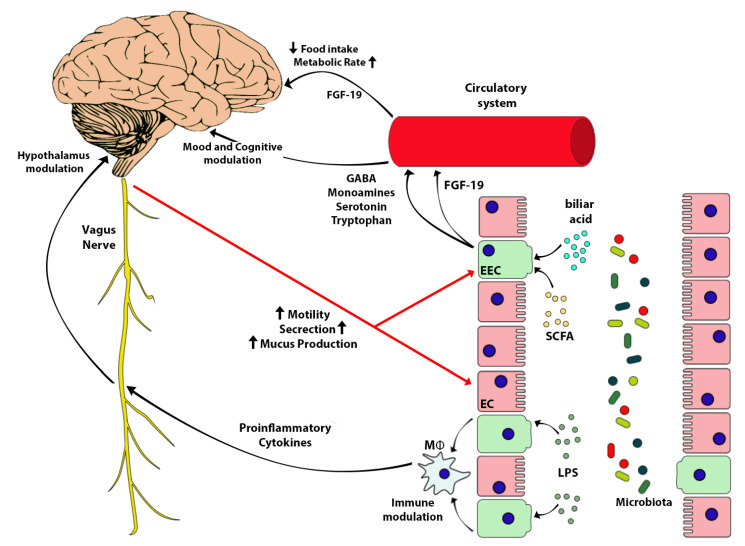
Brain/Gut axis as interaction between microbiota, gut cells, and peripheral and central nervous system in controlling food intake, metabolism, and the cognitive and circulatory systems. Gut microbiota releases chemical mediators and metabolites, such as SCFA and bile acids that stimulate enterochromaffin cells to secrete FGF-19, GABA, monoamines, serotonin and tryptophan. FGF-19 reduces food intake and increases metabolic rate. GABA, monoamines, fibroblast growth factor 19; EEC, enterochromaffin cell; EC, enteric cell; SCFA, short chain fatty acid; Mϕ, macrophage.

**Table 1 nutrients-13-00613-t001:** Adipokines, release and function.

Adipokines	Site of Release	Stimulated by	Function	Reference
Adiponectin	PVAT	Plasma concentration is inversely correlated with BMI and visceral fat	↑ insulin-mediated glucose uptake in skeletal muscle ↑ liver insulin sensitivity ↑ endothelium vasoactivity ↑ glucose production by gluconeogenesis	[52,53,54,55]
Leptin	PVAT	Plasma levels ↑ with fat stores	Regulator of energy intake Adaptation to fasting ↑ oxidative cellular stress by long exposure ↓ NO endothelial availability by long exposure	[58,59,60,61]

Abbreviation: PVAT, perivascular adipose tissue.

**Table 2 nutrients-13-00613-t002:** Gut hormones, release and function.

Gut Hormones	Site of release	Stimulated by	Function	Reference
GLP-1	Intestine L cells	Secreted after meal	↑ insulin secretion ↑ pancreatic beta cells survival and proliferation ↑ pancreatic alfa cells inhibition of glucagon secretion ↓ lipid secretion ↓ glucose production by liver ↓ gastric motility ↑ sense of satiety ↓ food ingestion ↓ oxidative stress/platelet aggregation ↑ insulin stimulated vasodilator reactivity	[106,107]
GIP	Intestine K cells	Secreted after meal	↑ insulin secretion ↑ beta cells proliferation ↓ beta cell apoptosis	[107]
OMX	Intestine L cells	Secreted after meal	↑ insulin secretion ↓ food ingestion	[108,109,110]
Ghrelin	Gastric P/D1 cells	Secreted during fasting	↑ food ingestion ↑ GLP-1 secretory ↑ NO endothelial availability	[111,112,113]
Obestatin	Gastric P/D1 cells	Secreted during fasting	↑ food ingestion ↑ NO endothelial availability ↑ pancreatic beta cells survival and proliferation ↓ lipolysis in adipose tissue	[114,115,116,117,118]
PYY	Intestine L cells	Secreted after meal	↓ gastric motility/↓ intestinal motility ↑ sense of satiety ↑ pancreatic secretion	[119,120,121,122]
INSL-5	Intestine L cells	Secreted during fasting	↑ food ingestion	[123,124]
CKK	Intestine I cells	Secreted after meal	↑ gallbladder contraction inhibition of gastric acid secretion ↑ insulin secretion ↑ pancreatic beta cells survival and proliferation ↑ sense of satiety	[95,125,126,127,128,129]

Abbreviations: GLP-1, glucagon like peptide-1; GIP, glucose-dependent insulinotropic peptide; OMX, oxyntomodulin; PYY, peptide tyrosine tyrosine; INSL-5, insulin-like peptide 5; CKK, cholecystokinin.

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
