# Peer review of "Association of Gut Hormones and Microbiota with Vascular Dysfunction in Obesity"

_nutrients, 2021, doi:10.3390/nu13020613_

Round 1

Reviewer 1 Report

Rovella V. and coworkers present the manuscript entitled “Obesity Related Vascular Dysfunction: Role of Gut and Gut Hormones” which harvest the last studies relating the influence of gut on the energy balance and its influence on the obese individuals.

The review is well written and of high interest, however some points must be addressed:

Tittle of the manuscript promise a compilation of studies relating the vascular dysfunction and gut influence over the energy metabolism, however, the manuscript does not relate the vascular effects of gut hormones, incretins or gut microbiota on the vascular function. I suggest to change the title, or alternatively, go deeper into the vascular effects of several gut factor influencing endothelial function and effects on several vascular beds. In the first part of the manuscript there are described the endothelial factors that influence vascular function, but there are not mentioned the effect of gut on them.

Endothelial dysfunction is defined as imbalance between vasodilators (not only NO) and vasoconstrictors factors (not only ET1). NO and ET1 are the dual endothelial factors induced by insulin, however there are many more endothelial factors that have vasodilator and vasoconstrictor effects and contribute to endothelial and vascular function. Please, correct, accordingly.

Authors do not mention that ET1 has both effects vasoconstrictor and also vasodilator effects.

Vascular effects of insulin have been demonstrated in several vascular beds, including coronary arteries, please, include. Furthermore, coronary arteries show a diminished effects of vasoconstrictor insulin-dependent MAPK pathway in order to protect this vital vascular bed. These studies must be included in the review.

Regarding the metabolic effects of GLP1 and analogues, it must be mentioned the last described central hypothalamic effects favoring metabolic status in obese individuals.

Introduce the abbreviations: Agouti related peptide (AgrP), and neuropeptide Y (NPY). Also briefly mention their orexigenic or anorexigenic effects, for better understanding of not expert in central energy metabolism. Also introduce PYY significance.

Effects of gut microbiota on the other endothelial factors different than NO, for example ET1, prostaciclin, tromboxane A2... If there are not studies, indicate in the manuscript.

ANS is only efferent. Gut communicate with CNS through afferent sensorial nerves different of ANS, nutrients and several gut hormones mentioned alongside the manuscript. Authors mention the afferent vagal signals as part of ANS, but this statement is not correct. Please correct this point.

References are numerally alongside the manuscript, however, in the bibliography reference are not numerated hindering the localization of the cites.

Author Response

Comment 1:

Tittle of the manuscript promise a compilation of studies relating the vascular dysfunction and gut influence over the energy metabolism, however, the manuscript does not relate the vascular effects of gut hormones, incretins or gut microbiota on the vascular function. I suggest to change the title, or alternatively, go deeper into the vascular effects of several gut factor influencing endothelial function and effects on several vascular beds. In the first part of the manuscript there are described the endothelial factors that influence vascular function, but there are not mentioned the effect of gut on them.

Answer 2: We are grateful to the reviewer for this comment. As suggested we now modified the titled as: “Association of Gut Hormones and Microbiota with Vascular Dysfunction in Obesity

Comment 2:

Endothelial dysfunction is defined as imbalance between vasodilators (not only NO) and vasoconstrictors factors (not only ET1). NO and ET1 are the dual endothelial factors induced by insulin, however there are many more endothelial factors that have vasodilator and vasoconstrictor effects and contribute to endothelial and vascular function. Please, correct, accordingly.

Answer 2: We thank the reviewer for this insight comment. We revised the text by adding description on other factors involved in the insulin-vascular dysfunction pathway (page 3, paragraph 2, reference 15).

Comment 3:

Authors do not mention that ET1 has both effects vasoconstrictor and also vasodilator effects.

Answer 3: We clarified this important point by adding a sentence and related reference in the text (page 3, paragraph 6, reference 25).

Comment 4:

Vascular effects of insulin have been demonstrated in several vascular beds, including coronary arteries, please, include. Furthermore, coronary arteries show a diminished effects of vasoconstrictor insulin-dependent MAPK pathway in order to protect this vital vascular bed. These studies must be included in the review.

Answer: In agreement with reviewer, we clarified better the role of insulin in the vascular bed and specifically in coronary arteries (page 4, paragraph 3, reference 42)

Comment 5:

Regarding the metabolic effects of GLP1 and analogues, it must be mentioned the last described central hypothalamic effects favoring metabolic status in obese individuals.

Answer 5: In agreement with the reviewer we added a sentence to clarify this point and the related reference (page 9, paragraph 1, reference 114)

Comment 6:

Introduce the abbreviations: Agouti related peptide (AgrP), and neuropeptide Y (NPY). Also briefly mention their orexigenic or anorexigenic effects, for better understanding of not expert in central energy metabolism. Also introduce PYY significance.

Answer 6: As requested by the reviewer we did introduce the abbreviation in the manuscript. Moreover, we did report their orexigenic and anorexigenic effects in the text of manuscript (page 9, section 5.3 paragraph 1).

Comment 7: Effects of gut microbiota on the other endothelial factors different than NO, for example ET1, prostaciclin, tromboxane A2... If there are not studies, indicate in the manuscript.

Answer 7: As requested by reviewer we did check for these studies. However, at the best of our knowledge, we did find only study that involved NO as mediator, even if ET1 and other factors were taken into consideration. We did report this statement on the manuscript (page 19 section 9 paragraph 4)

Comment 8:

ANS is only efferent. Gut communicate with CNS through afferent sensorial nerves different of ANS, nutrients and several gut hormones mentioned alongside the manuscript. Authors mention the afferent vagal signals as part of ANS, but this statement is not correct. Please correct this point.

Aswer 8: In agreement with the reviewer we did correct this point as “The afferent sensorial nerves are activated by gastric mechanoreceptors triggered by wall distension and specific nutrients like SCFA and others stimulating gut hormones releas. All these afferent….” (page 19, section 10, paragraph 2).

Comment 9:

References are numerally alongside the manuscript, however, in the bibliography reference are not numerated hindering the localization of the cites.

Answer 9: We apologize for this inconvenience. Now this problem has been solved.

Reviewer 2 Report

The introduction of the review feels disjointed and the first and second paragraph are essentially making the same point. The authors need to be more succinct with the points they wish to express. Further, paragraphs 1, 2 and 3 are not different paragraphs but rather a continuation of the same research idea. The key and novel aspect is the role of the gut flora and gut hormones, however, the authors hardly comment on this in the introduction and as such they need to provide more insightful information on this aspect in order to set up the review.

The opening of the review starts with endothelial dysfunction and obesity. Which seems out of order as we first need to understand the condition of obesity and then how obesity leads to endothelial dysfunction. Further, are the authors talking about obesity with the associated pathologies of hypertension, insulin resistance (diabetes), hyperlipidemia? The lack of detail and specifics limits the review and its application to vascular function and the role of the gut.

Further, the endothelial dysfunction section reflects everything (in limited detail) from signaling issues on the endothelial with references to hypertension and then limited information on atherosclerosis, which is a time dependent process.  The authors then begin to provide information on how vascular dysfunction is evidence in obese humans, however, the clarity of this information is limited by the mixed conditions associated with obesity with the authors providing information from obese patients with diabetes and then later on metabolic syndrome. As such, the authors need to provide a more focused and succinct endothelial (and) endothelial-independent section.  Furthermore, it is unclear as to why the authors have also examined, in a separate section (when it is even mentioned in section 3),  endothelial dysfunction with metabolic syndrome (section 4). The authors then move into adipose tissue, insulin resistance and endothelial dysfunction (section 5). Up to now, all of this has been extensively reviewed many times and as such the current review limited information to past reviews on this topic. Indeed, there are more extensive and detail reviews focused on PVAT and endothelial dysfunction. The novel aspects, as to what the reader was expecting from the title and opening introduction, was the role of the gut, which does not come into section 6. This section the describes various gut hormones  and how they regulate glucose control etc and there is intermittent description as to how certain gut hormones  may modulate vascular function. However, this is limited and should be in more detail and a specific subsection once the gut hormones etc have been described. Indeed, there is a specific section (10) on this yet it is unclear as to why the authors reported some of this information in section 6, and it is also unclear as to why section 10 is limited in specific insights as to how the microbiome directly impacts vascular function. The information provided with regard to the gut microbiome is limited and should be accompanied by pre-clinical (animal studies) that have performed microbiome transplants and examined vascular function as direct evidence of this cross-talk.

The authors in section 10 superficially talk about far too many potential modulates rather than focusing on some of the most important and providing a more detailed and mechanistic data. This is another example of the lack of focus of the review.

The authors also do not disclose whether it is the gut hormones alone, or the excess adipose tissue (and their secreted products), or both that are negatively modulating vascular function. This is an important question.

Lastly, there are numerous grammatic and sentence structure issues, new paragraphs are inserted when the information is just a continuation.

Author Response

Comment 1:

The introduction of the review feels disjointed and the first and second paragraph are essentially making the same point. The authors need to be more succinct with the points they wish to express. Further, paragraphs 1, 2 and 3 are not different paragraphs but rather a continuation of the same research idea. The key and novel aspect is the role of the gut flora and gut hormones, however, the authors hardly comment on this in the introduction and as such they need to provide more insightful information on this aspect in order to set up the review.

Answer 1: We thank the reviewer for this insight comment. As per reviewer suggestion the introduction and the first part of the manuscript have been revised to make it more fluent and to provide more novelties on gut flora and hormones in order to insert the key argument of the paper (pages 1 and 2, paragraph 1).

Comment 2:

The opening of the review starts with endothelial dysfunction and obesity. Which seems out of order as we first need to understand the condition of obesity and then how obesity leads to endothelial dysfunction. Further, are the authors talking about obesity with the associated pathologies of hypertension, insulin resistance (diabetes), hyperlipidemia? The lack of detail and specifics limits the review and its application to vascular function and the role of the gut.

Answer 2: In agreement with the reviewer comment we now added a brief section (# 3) with the scope to make a better flow in our manuscript and with the aim to link obesity with endothelial dysfunction, as requested.

Comment 3:

Further, the endothelial dysfunction section reflects everything (in limited detail) from signaling issues on the endothelial with references to hypertension and then limited information on atherosclerosis, which is a time dependent process.  The authors then begin to provide information on how vascular dysfunction is evidence in obese humans, however, the clarity of this information is limited by the mixed conditions associated with obesity with the authors providing information from obese patients with diabetes and then later on metabolic syndrome. As such, the authors need to provide a more focused and succinct endothelial (and) endothelial-independent section.  Furthermore, it is unclear as to why the authors have also examined, in a separate section (when it is even mentioned in section 3),  endothelial dysfunction with metabolic syndrome (section 4). The authors then move into adipose tissue, insulin resistance and endothelial dysfunction (section 5). Up to now, all of this has been extensively reviewed many times and as such the current review limited information to past reviews on this topic. Indeed, there are more extensive and detail reviews focused on PVAT and endothelial dysfunction. The novel aspects, as to what the reader was expecting from the title and opening introduction, was the role of the gut, which does not come into section 6. This section the describes various gut hormones  and how they regulate glucose control etc and there is intermittent description as to how certain gut hormones  may modulate vascular function. However, this is limited and should be in more detail and a specific subsection once the gut hormones etc have been described. Indeed, there is a specific section (10) on this yet it is unclear as to why the authors reported some of this information in section 6, and it is also unclear as to why section 10 is limited in specific insights as to how the microbiome directly impacts vascular function. The information provided with regard to the gut microbiome is limited and should be accompanied by pre-clinical (animal studies) that have performed microbiome transplants and examined vascular function as direct evidence of this cross-talk.

Answer 3: In agreement with the whole comment of the reviewer, we now extensively revised these sections of the manuscript.  We condensed the paragraph on endothelial dysfunction, and included in one section the discussion regarding obesity and metabolic syndrome by rewriting this paragraph. We did focus more, as suggested, on PVAT and on the role of gut. We now removed description of gut hormones from page 6. We included a paragraph introducing the pre-clinical studies performed microbiome transplants and examining vascular function, as requested.

Comment 4:

The authors in section 10 superficially talk about far too many potential modulates rather than focusing on some of the most important and providing a more detailed and mechanistic data. This is another example of the lack of focus of the review.

Answer 4: We agree with the reviewer and then we reorganized the section 10 (now section 9) in order to give to it a better flow, and to provide a better understanding, even on the mechanistic processes regulating these organic function.

Comment 5:

The authors also do not disclose whether it is the gut hormones alone, or the excess adipose tissue (and their secreted products), or both that are negatively modulating vascular function. This is an important question.

Answer 5: In agreement with the reviewer, we did include this important point in the conclusions section to remark it (page 22, paragraph 2; page 23 paragraph 1)

Comment 6:

Lastly, there are numerous grammatic and sentence structure issues, new paragraphs are inserted when the information is just a continuation.

Answer 6: We thank the reviewer and now the article has been extensively revised also for grammatic issues.

Reviewer 3 Report

Rovella and coworkers provide a detailed and very broad review of recent literature describing the complex interplay between obesity and endothelial dysfunction, conditions that were linked to incretin hormone function, microbiome and altered energy metabolism. It is comprehensive, nicely written, intelligible, highly topical and therefore certainly deserves publication in Nutrients. In particular the characteristics of endothelial dysfunction in obesity and the metabolic syndrome are nicely outlined. Here, I have a number of comments, which may be instrumental to improve this review manuscript.

Major comments:

  1. Please re-write the abstract to be more focussed. As it stands, the abstract does not show a red thread and deals with a multitude of medical terms that are not logically connected. Also the introduction should be expanded and more focussed.
  2. The references should be numbered but they are not in the reference list. Why? This makes it hard for me to read the manuscript along with the references.
  3. Is the vasodilator activity of insulin action on endothelial cells part of its anabolic activity? What other anabolic roles does insulin exert. Please discuss in more detail on page 4.
  4. I recommend to summarize the role of adipokines in a table. In that way, it is possible to reduce the text and avoid the listing of these functions that make the manuscript harder to follow.
  5. The manuscript would strongly benefit from a paragraph descirbing the microbiota-dependent regulation of gut hormones and refer to relevant literature (e.g. Wichmann A et al., Cell Host Microbe, 2013; Ying Shiuan Lee et al., Mol Metab., 2016; Greiner TU & Bäckhed F, Mol Metab., 2016; Grasset E et al., Cell Metab., 2019; Martchenko SE et al., Diabetes, 2020).
  6. When dealing with the microbiota’s role in promoting vascular dysfunction, vascular inflammation and cardiovascular disease, there is additional relevant literature that should be cited (Karbach SH et al., J Am Heart Assoc., 2016; Kiouptsi K et al., mBio, 2019; Kiouptsi K et al., Gut Microbes, 2020).

Minor comments:

  1. Please stick to scientific language and avoid expressions like “terrible pandemic”.
  2. The authors write: “Endothelium is an inner lining of the vessels and is composed by a singular layer epithelial cells surrounded by smooth vascular muscle cells”. Do they mean that …composed by a singular layer of endothelial cells surrounded by…, which interact to …? Please check.
  3. When dealing with perivascular adipose tissue, I recommend to refer to the work of Xia N et al, ATVB, 2016 describing the uncoupling of eNOS in obesity.
  4. When describing the regulation of physiological activities by the gut microbiota on page 11, I recommend to also include the characteristic influence of the microbiota on the formation of capillary networks in the small intestinal mucosa (Reinhardt C, Nature, 2012). This is relevant since it can assist nutrient uptake.
  5. On page 15, when dealing with the role of microbiota-derived LPS on metabolic inflammation, the landmark paper by Caesar R et al., Gut, 2012 should be cited.

Round 2

Reviewer 1 Report

Authors have successfully addressed all my concerns.

Reviewer 2 Report

I thank the authors for considering my comments. There are a number of grammatical issues, that clearly become evident in the abstract. References are missing when the authors state "several studies". These factors should be addressed